# Azole-induced cell wall carbohydrate patches kill *Aspergillus fumigatus*

Bernadette Geißel[1], Veronika Loiko[1], Isabel Klugherz[1], Zhaojun Zhu[1], Nikola Wagener[2], Oliver Kurzai[3,4], Cees A.M.J.J. van den Hondel[5] & Johannes Wagener[1,3,4]

Azole antifungals inhibit the fungal ergosterol biosynthesis pathway, resulting in either growth inhibition or killing of the pathogen, depending on the species. Here we report that azoles have an initial growth-inhibitory (fungistatic) activity against the pathogen *Aspergillus fumigatus* that can be separated from the succeeding fungicidal effects. At a later stage, the cell wall salvage system is induced. This correlates with successive cell integrity loss and death of hyphal compartments. Time-lapse fluorescence microscopy reveals excessive synthesis of cell wall carbohydrates at defined spots along the hyphae, leading to formation of membrane invaginations and eventually rupture of the plasma membrane. Inhibition of β-1,3-glucan synthesis reduces the formation of cell wall carbohydrate patches and delays cell integrity failure and fungal death. We propose that azole antifungals exert their fungicidal activity by triggering synthesis of cell wall carbohydrate patches that penetrate the plasma membrane, thereby killing the fungus. The elucidated mechanism may be potentially exploited as a novel approach for azole susceptibility testing.

[1] Max von Pettenkofer-Institut für Hygiene und Medizinische Mikrobiologie, Medizinische Fakultät, LMU München, Pettenkoferstraße 9a, 80336 Munich, Germany. [2] Zell- und Entwicklungsbiologie, Department Biologie II, LMU München, Großhaderner Straße 2, 82152 Planegg-Martinsried, Germany. [3] Institut für Hygiene und Mikrobiologie, Julius-Maximilians-Universität Würzburg, Josef-Schneider-Straße 2, 97080 Würzburg, Germany. [4] National Reference Center for Invasive Fungal Infections (NRZMyk), Leibniz-Institut für Naturstoff-Forschung und Infektionsbiologie, Hans-Knöll-Institut, Adolf-Reichwein-Straße 23, 07745 Jena, Germany. [5] Molecular Microbiology and Biotechnology, Institute of Biology Leiden, Leiden University, Sylviusweg 72, 2333 BE Leiden, The Netherlands. Correspondence and requests for materials should be addressed to J.W. (email: j.wagener@hygiene.uni-wuerzburg.de)

The antifungal activity of an azole (benzimidazole) was first described in 1944[1]. Since then, azoles have come a long way, nowadays representing one of the most important drug classes for the control of plant diseases caused by fungi as well as for the treatment of fungal infections in human and veterinarian medicine. Azoles can be categorized in two groups, the imidazole derivatives (e.g., ketoconazole and miconazole) and the newer triazole derivatives (e.g., fluconazole and voriconazole). Both groups have in common that they interfere with ergosterol biosynthesis. Ergosterol is the primary sterol in fungal membranes and presumably contributes to membrane fluidity and function[2,3]. Azoles act by directly inhibiting the lanosterol 14α-demethylase (CYP51), a key enzyme in the ergosterol biosynthesis pathway, which catalyzes demethylation of the intermediates eburicol or zymosterol at position C-14. As a consequence, the fungal cells suffer from depletion of ergosterol and, according to some authors, from the accumulation of toxic sterol precursors (14α-methyl sterols)[2,4].

While the interaction of azoles and their target enzymes have extensively been analyzed on a molecular and structural level (reviewed in ref.[5]), surprisingly little is known about the physiological consequences of CYP51 inhibition on the fungal cell biology. Based on several studies performed in the early 1980s, it was hypothesized that ergosterol depletion affects fungal viability and growth in various ways: by increasing the permeability of membranes, decreasing or increasing the activity of membrane-bound enzymes in the plasma membrane and mitochondria, stimulating uncoordinated chitin synthesis or interfering with fatty acid synthesis (reviewed in refs.[2,6]). More recently, it has been proposed that azoles trigger the production of reactive oxygen and nitrogen species in certain fungi[7–10]. Some of the effects were potentially attributed to additional, CYP51-independend activities of the imidazole derivatives used at this time, such as direct binding to lipids[6,11]. Undoubtedly, azoles have divergent effects depending on the fungal species. They exert generally a fungistatic activity against yeasts, e.g., *Candida* spp., while being fungicidal for certain medically important molds, e.g., *Aspergillus* spp[3,12]. The nature of the fungicidal effect on *Aspergillus* species remained essentially unexplained.

Here, we provide an explanation on how azoles kill the major fungal pathogen *Aspergillus fumigatus*. This mold is the primary cause of invasive aspergillosis, a severe and life-threatening infection with an estimated global incidence of 200,000 human cases per year and a mortality rate of 30 to 95 %[13,14]. The triazole voriconazole is highly effective against *A. fumigatus* and recommended as first-line treatment. Here, we show that the fungicidal activity of voriconazole against *A. fumigatus* is linked to azole-induced cell wall remodeling defects, which cause cell wall stress, bending of the plasma membranes to the inside, cell wall integrity failure, and death. The death is associated with heterogeneous phenotypes, which include expulsive release of cytoplasm, mitochondrial fragmentation, and mitochondrial lysis. Further, we demonstrate that the fungicidal effect is a distinct mechanism that can be separated from the solely fungistatic activity of the azoles. Our findings are linked to azole-mediated inhibition of CYP51 because we were able to reproduce all observations by genetically depleting the lanosterol 14α-demethylase. Moreover, analysis of clinical *A. fumigatus* isolates suggests a potential application in routine diagnostics for detecting clinically relevant azole resistance.

## Results

### Manifestations of azole-induced fungal death is heterogeneous.
In viable cells, mitochondria form tubular and highly dynamic networks. We exposed *A. fumigatus* wild-type hyphae expressing a mitochondria-targeted green fluorescent protein (GFP) to

fungicidal concentrations of voriconazole. Examination of static images of samples fixed after approximately 12 h voriconazole exposure revealed a mixed picture. Several hyphae did not show any fluorescence signal. Some hyphae presented a tubular, others a highly fragmented mitochondrial morphology. Occasionally, we observed hyphae with homogeneous cytosolic GFP fluorescence and GFP positive vesicles outside of hyphae. To understand this heterogeneity, we followed the fate of individual *Aspergillus* hyphae exposed to voriconazole with microscopy over time. As shown in Fig. 1a–c and Supplementary Movies 1–6, we observed essentially three manifestations of voriconazole-induced fungal death: First, sudden expulsive release of cytoplasm (Fig. 1a and Supplementary Movies 1 and 2). Second, arrest of mitochondrial dynamics combined with mitochondrial fragmentation (Fig. 1b and Supplementary Movies 3 and 4). Third, intracellular lysis of mitochondria (Fig. 1c and Supplementary Movies 5 and 6). As expected, voriconazole caused a significant and concentration-dependent growth repression within 1 h after addition. However, the hyphae always survived for at least 2.5–3 h, independent of the applied azole concentration (Supplementary Movies 1–6). As shown in Fig. 1d, the frequency of the individual manifestations of death was directly related to the applied azole concentration. At lower concentrations we observed primarily the expulsive release of the cytoplasm, at higher concentrations mitochondria preferentially lyse, releasing their mitochondria-targeted GFP to the cytosol (see Supplementary Movies 7–10). Importantly, careful examination of the movies revealed a sudden subtle shrinking of the hyphae immediately before the occurrence of mitochondrial fragmentation or lysis (Supplementary Movies 1–10). This indicates a cell integrity failure followed by lowering and dissipation of the intracellular pressure preceding all three manifestations of death.

### Azole-induced stress activates the fungal cell-wall salvage system.
In *A. fumigatus*, maintenance of the cell wall integrity (CWI) relies on a conserved fungal stress signaling pathway (reviewed in ref.[15]). As our time-lapse microscopy data indicated cell integrity failure, we analyzed the activation state of this pathway upon exposure of *A. fumigatus* to voriconazole over time. To this end, we applied a novel reporter construct where the firefly luciferase is placed under the control of the *Aspergillus niger agsA* promoter, which is readily induced by the cell wall salvage system upon stress[16]. Interestingly, the reporter was strongly induced after a two to 3 h lag phase following azole addition (Fig. 1e). In agreement with our previous results, the voriconazole concentration did not influence the time to onset, but correlated with the strength of induction. Taken together, this indicates that azoles trigger activation of the cell wall salvage system at the same time when the hyphae begin to die.

### Hyphal septa improve survival of azole-treated *Aspergillus* hyphae.
We have recently shown that hyphal septa are essential for survival of *A. fumigatus* exposed to echinocandin antifungals[17]. Echinocandins inhibit biosynthesis of the cell wall carbohydrate β-1,3-glucan. As a consequence, hyphal cell walls occasionally rupture, thereby dooming the affected hyphal compartment to death. We speculated that septa could have a similar protective role for survival of *A. fumigatus* hyphae exposed to azoles. Hyphae of wild-type and of an *Aspergillus* mutant that is unable to form septa (Δ*rho4*)[18] as well as the complemented mutant (*rho4*) were exposed to voriconazole. To discriminate viable from dead hyphae, strains were used that constitutively express cytosolic GFP. After 5 h, hyphae were stained with trypan blue to quench the GFP signal in lysed compartments and analyzed with a fluorescence microscope (Fig. 1f). In agreement with

our hypothesis, the amount of viable microcolonies (herein defined as microscopic colonies consisting of hyphae that originate from single conidia) of the Δrho4 mutant was drastically reduced compared to wild-type and the complemented mutant. Remarkably, the number of fully viable microcolonies of the wild-type and the complemented mutant were in the same range as those of the Δrho4 mutant. Although this clearly shows that septa contribute to survival of *A. fumigatus* challenged by azoles, the minimal concentration required to inhibit growth of the Δrho4 mutant is similar to that of wild-type[18]. In addition, prolonged

voriconazole exposure of the wild-type (e.g., >24 h) caused death of essentially all microcolonies (not shown). This demonstrates that hyphal septa can extend the survival time but not prevent death caused by azole antifungals.

**Azoles trigger the formation of cell wall carbohydrate patches.** As shown in Fig. 1f, the trypan blue dye strongly stained several undefined patch-like structures within the dead and living voriconazole-exposed hyphae. These structures were not present

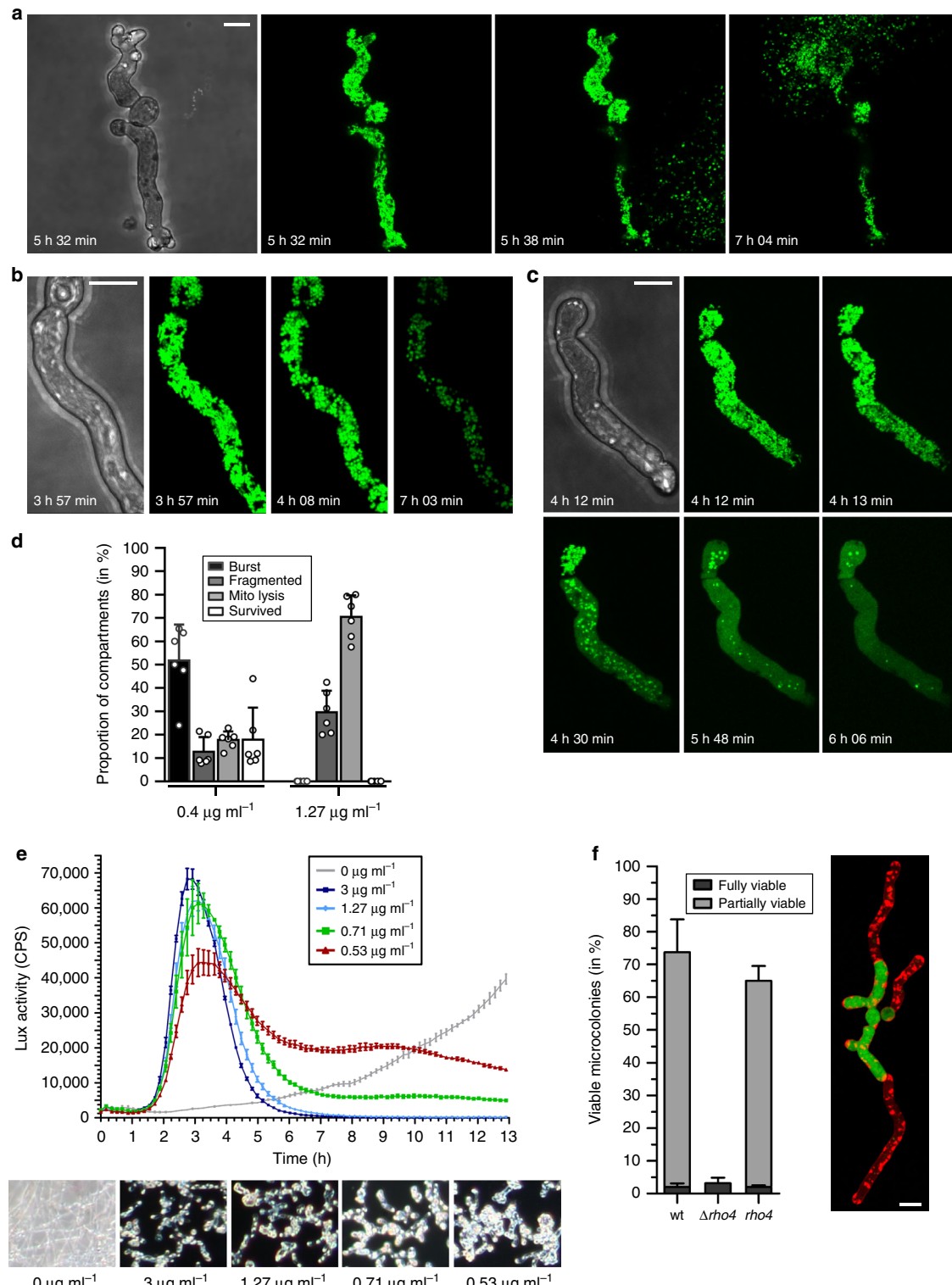

in hyphae not exposed to azoles (Supplementary Fig. 1). Only recently, it was reported that trypan blue can label cell wall carbohydrates chitin and glucan[19]. To validate and eventually discriminate these potential two carbohydrates, we stained hyphae raised under similar conditions with the chitin-specific dye calcofluor white and the glucan-specific dye aniline blue. Both dyes strongly accumulated in the patch-like structures, thereby indicating that they contain large amounts of both chitin and glucan (Fig. 2a, b). Intriguingly, the formation of the cell wall carbohydrate deposits strongly correlated with the induction of the cell wall salvage system (Fig. 2c).

All experiments so far have been performed with hyphae. However, it has been reported that azoles also act on conidia of *A. fumigatus*. After exposure to fungicidal concentrations of an azole conidia lose the ability to germinate[12]. We were wondering if carbohydrate patches could also be observed in conidia when treated with azoles, which is indeed the case as shown in Fig. 2d. The conidia accumulated carbohydrate patches that fill up to one quarter of the conidial volume. Finally, the conidia died similar to azole-exposed hyphae (Fig. 2d).

**Repression of CYP51 phenocopies azole-treated wild-type.** Several antifungal activities reported for azole derivatives in earlier studies are probably not linked to inhibition of the target enzyme lanosterol 14α-demethylase[6,11]. We, therefore, questioned whether the effects we observed in our study are related to inhibition of the lanosterol 14α-demethylase by voriconazole or constitute target-independent effects. To this end, we constructed a mutant that allowed us to specifically address CYP51-dependent effects by genetically depleting the target enzyme. *A. fumigatus* encodes two functionally redundant CYP51 genes, *cyp51A* and *cyp51B*. To conditionally repress CYP51, a mutant was constructed where we replaced the endogenous promoter of *cyp51A* with a doxycycline-inducible Tet-On promoter and subsequently deleted *cyp51B*. The resulting mutant, $cyp51A_{tetOn}\Delta cyp51B$, had no apparent growth phenotype compared to wild-type under induced conditions, but was non-viable under repressed conditions. To visualize the mitochondrial morphology, the conditional CYP51 mutant was additionally transformed with a construct for expression of mitochondria-targeted GFP. Repression of CYP51 caused an initial growth inhibition followed by the three different manifestations of fungal death, which we also found after treatment of wild-type with voriconazole: that is expulsive release of mitochondria (cytoplasm) (Fig. 3a), fragmentation of the tubular mitochondrial network (Fig. 3b) and mitochondrial lysis (Fig. 3c). In addition, we observed excessive synthesis of cell wall carbohydrates at defined spots, very similar to the chitin and

glucan accumulation observed with wild-type exposed to voriconazole (Fig. 3d). This clearly demonstrates that the effects of voriconazole we describe above are exclusively attributed to inhibition of the lanosterol 14α-demethylase.

**Cell wall carbohydrate patches invaginate the plasma membrane.** The microscopic examination of the chitin and glucan-stained voriconazole-exposed hyphae and conidia suggested a localization of the patches within the hyphae or conidia. This could mean that either irregular cell wall synthesis occurs within chitin and glucan synthase loaded secretory vesicles in the cytoplasm[20] or that excessive synthesis occurs at defined patches at the cell surface, thereby forcing the plasma membrane into the hyphal body. To clarify the exact topology of the chitin and glucan deposits, we expressed a GFP-tagged Wsc1 cell wall stress sensor in *A. fumigatus*. This type I transmembrane protein is evenly distributed at the plasma membrane under normal growth conditions[18]. Time-lapse microscopy of Wsc1-GFP-expressing hyphae treated with voriconazole demonstrated the formation of remarkable invaginations of the plasma membrane up to 3 μm in size. These invaginations co-localized with sites where the cell wall carbohydrate patches were formed (Fig. 4 and Supplementary Movies 11–13). Interestingly, the fluorescence intensity of Wsc1-GFP significantly increases at sites of invaginated membranes (fluorescence heatmap, Fig. 4). Very similar results were also obtained with a different GFP-tagged cell wall stress sensor (MidA-GFP)[18]. This demonstrates that transport vesicles loaded with stress sensors and, presumably, other membrane proteins are continuously delivered to the sites of excessive cell wall biogenesis under azole stress.

**Attenuated fungicidal activity in respiratory chain mutants.** The time-lapse microscopy results indicated that the fungistatic effect of the azoles precedes the fungicidal activity by several hours. This suggested that the fungicidal activity is a discrete effect, which occurs in parallel or builds on top of the fungistatic activity. We and others recently reported a link between mitochondrial dysfunction and azole susceptibility of *A. fumigatus*[21,22]. In the sequel of these studies, we constructed mutants that are affected in the mitochondrial respiratory pathway. Specifically, we replaced the endogenous promoters of *rip1* (AFUA_5G10610), the gene encoding the Rieske iron-sulfur protein, a catalytic subunit of mitochondrial complex III, and of *cycA*, the gene encoding cytochrome C, with a doxycycline-inducible Tet-On promoter. Although *A. fumigatus* strictly depends on a functional respiratory chain, these mutants are viable under repressed conditions thanks to the alternative

**Fig. 1** Multiple manifestations of voriconazole-induced death are linked to cell wall integrity failure. **a–d** *A. fumigatus* wild-type conidia expressing mitochondria-targeted GFP were inoculated in Sabouraud medium and incubated at 37 °C. After 9 h, medium was supplemented with 0.4 μg ml$^{-1}$ (**a**), 1.27 μg ml$^{-1}$ (**b, c**) or the indicated amount (**d**) of voriconazole. The fate of individual hyphae was followed over time with confocal laser scanning microscopy. **a–c** Exemplary bright field and time-lapse GFP fluorescence images (green) of optical stacks covering the entire hyphae in focus are depicted. **d** Quantitative analysis of three voriconazole-induced fungal death manifestations. Approximately 160 hyphal compartments were analyzed per condition for 13 h. Bars represent means of the six individual data points, error bars indicate standard deviations. Shown are results representative of two independent time-lapse microscopy experiments per condition. **e** *A. fumigatus* conidia harboring a luciferase-based cell wall salvage reporter were inoculated in a 96-well plate in Sabouraud medium and incubated at 37 °C. After 7 h, luciferin and the indicated amount of voriconazole were added. Upper panel, luciferase activity over time after addition of voriconazole. Lower panel, exemplary microscopic dark-field images of hyphae after 17 h co-incubation. Data are representative of three independent experiments. **f** Conidia of wild-type, Δ*rho4* and *rho4* expressing cytosolic GFP were inoculated in Sabouraud medium. After 11 h incubation at 37 °C, medium was supplemented with 1.27 μg ml$^{-1}$ voriconazole. After 5 h co-incubation, hyphae were stained with trypan blue to quench the GFP signal in lysed compartments. The percentage of viable microcolonies was determined (graph). Data points represent means, the error bars indicate standard deviations. Data are representative of six independent blinded experiments. An exemplary overlay fluorescence image of optical stacks covering a partially viable wild-type hypha (green, GFP; red, trypan blue) is depicted on the left. **a–c, f** Bars represent 10 μm and are applicable to all subpanels

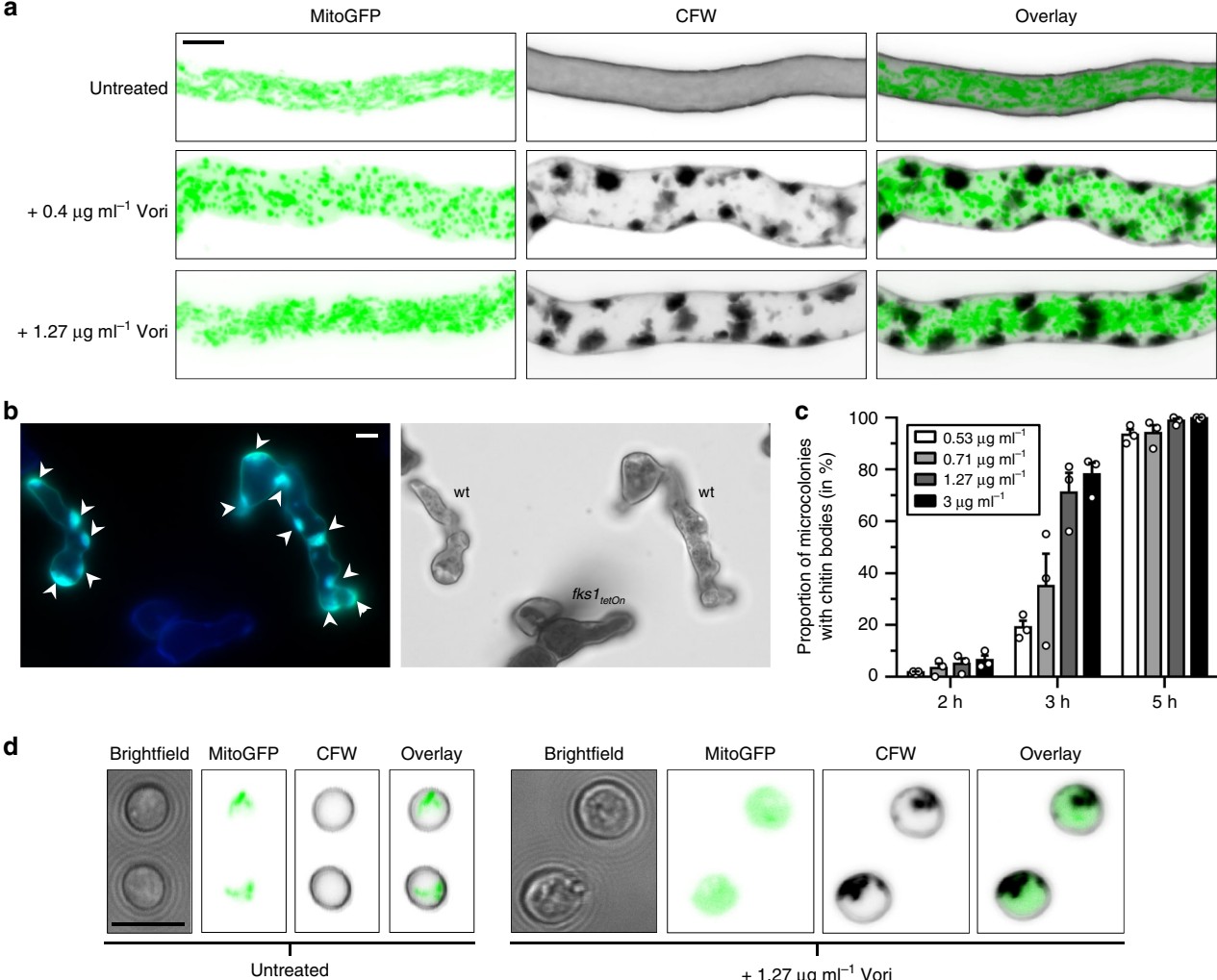

**Fig. 2** Inhibition of the lanosterol 14α-demethylase triggers excessive synthesis of cell wall carbohydrates at defined foci. **a–c** Conidia of *A. fumigatus* wild-type expressing mitochondria-targeted GFP (**a**), cytosolic GFP (**b**), or no GFP (**c**) were inoculated in Sabouraud medium on cover slips and incubated at 37 °C. The experiment depicted in **b** was additionally inoculated with conidia of a conditional β-1,3-glucan synthase mutant under repressive conditions (*fks1*$_{tetOn}$; staining control). **a** When indicated (+ Vori), medium was supplemented with 0.4 or 1.27 µg ml$^{-1}$ voriconazole 10 h after inoculation. After a total of 15 h incubation, hyphae were fixed, stained with calcofluor white and analyzed with a confocal laser scanning microscope. Depicted are representative images of optical stacks of mitochondria (GFP; left panels), chitin (calcofluor white; middle panels) and an overlay (right panels) that cover the entire hypha in focus. **b** After 8 h incubation, medium was supplemented with 1.27 µg ml$^{-1}$ voriconazole. After additional 9 h incubation at 37 °C, hyphae were fixed, stained with aniline blue and immediately analyzed with a fluorescence microscope. Left, glucan-specific (green) and nonspecific (*fks1*$_{tetOn}$; blue) aniline blue fluorescence. Right, bright field microscopy. Arrow heads indicate glucan patches. **c** After 7 h incubation, medium was supplemented with the indicated amount of voriconazole. After 2, 3, and 5 h co-incubation, samples were fixed and stained with calcofluor white. The percentage of microcolonies with chitin patches was determined with a fluorescence microscope and plotted in the depicted graph. Bars represent means of the indicated data points, error bars indicate standard deviations. Data are representative of three independent blinded experiments. **d** Wild-type conidia expressing mitochondria-targeted GFP were stained with calcofluor white, either directly (resting conidia; left panel) or after 45 h incubation at 37 °C in Sabouraud medium supplemented with 1.27 µg ml$^{-1}$ voriconazole (right panel). Depicted are representative bright field images (left), images of single GFP (middle left) and calcofluor white (middle right) fluorescence cross sections and an overlay of both (right). **a**, **b,** and **d** Bars represent 5 µm and are applicable to all subpanels

oxidase. This enzyme can catalyze the electron transfer from reduced ubiquinone directly to oxygen and thereby bypasses complex III and IV[23]. Both conditional mutants, *cycA*$_{tetOn}$ and *rip1*$_{tetOn}$, exhibited an unexpected and surprising phenotype under repressed conditions (Fig. 5). Disruption of complex III as well as downregulation of cytochrome C resulted in minimal growth of *Aspergillus* hyphae within voriconazole Etest inhibition zones (Fig. 5a). This is in marked contrast to wild-type or to *cycA*$_{tetOn}$ and *rip1*$_{tetOn}$ under induced conditions where the conidia die and the inhibition zones of Etests are typically void of any hyphal growth (Fig. 5a, compare Fig. 2d). At the same time macroscopic examination of the plates suggested a minimal inhibitory concentration for *cycA*$_{tetOn}$ and *rip1*$_{tetOn}$ under repressed conditions similar or even lower than that for wild-type (Fig. 5a). Very similar results were obtained using the broth microdilution method (Fig. 5b, c). These results indicate that voriconazole exerts a general fungistatic activity against *A. fumigatus* independent of the mitochondrial electron transport

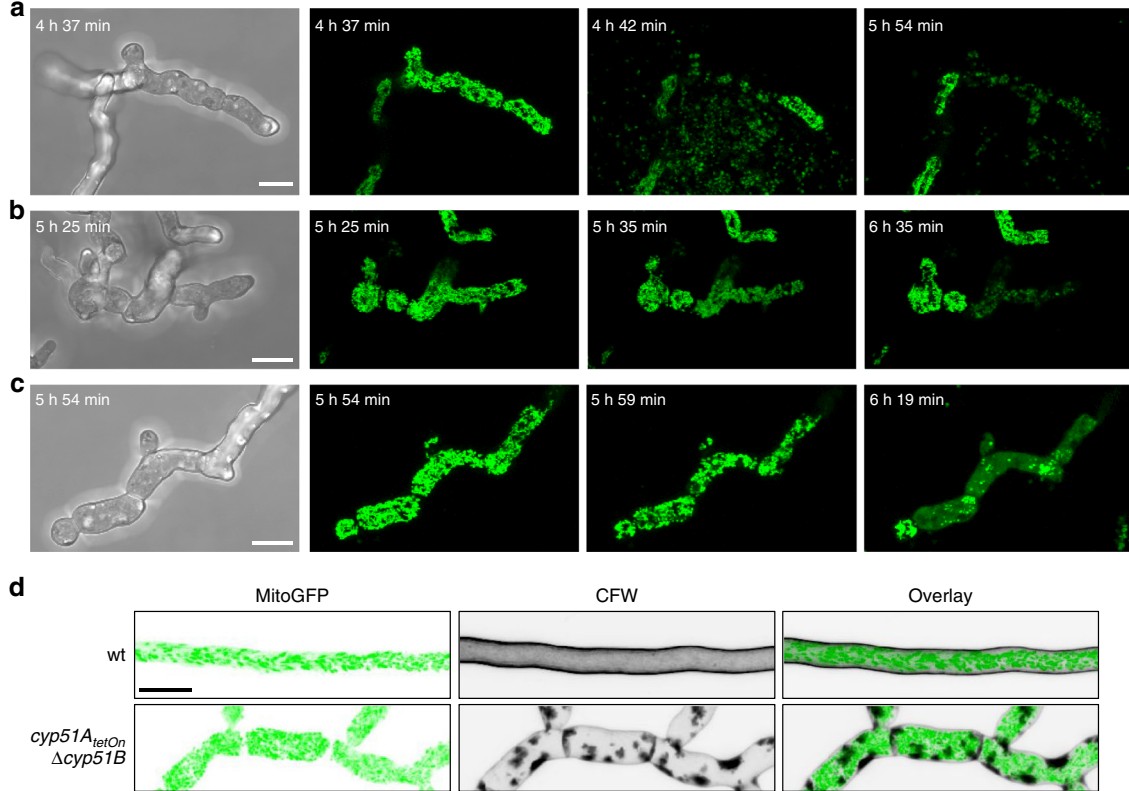

**Fig. 3** Azole-triggered cell death and synthesis of cell wall carbohydrate patches are linked to inhibition of the lanosterol 14α-demethylase. **a–d** Conidia of the conditional $cyp51A_{tetOn}\Delta cyp51B$ strain (**a– d**) and wild-type (**d**) that express mitochondria-targeted GFP were inoculated under induced conditions in Sabouraud medium supplemented with 15 µg ml⁻¹ doxycycline. After 9 h incubation at 37 °C, hyphae were shifted to repressive conditions by substitution of the medium without doxycycline. **a–c** The fate of individual hyphae was followed over time with a confocal laser scanning microscope. Depicted are exemplary bright field and time-lapse GFP fluorescence images (green) of optical stacks covering the entire hyphae in focus. **d** After 5 h incubation under repressive conditions, hyphae were fixed, stained with calcofluor white and subjected to confocal laser scanning microscopy. Depicted images represent optical stacks of GFP fluorescence (left panels), calcofluor white fluorescence (middle panels) and an overlay (right panels) that cover the entire hypha in focus. **a–d** Bars represent 10 µm and are applicable to all subpanels

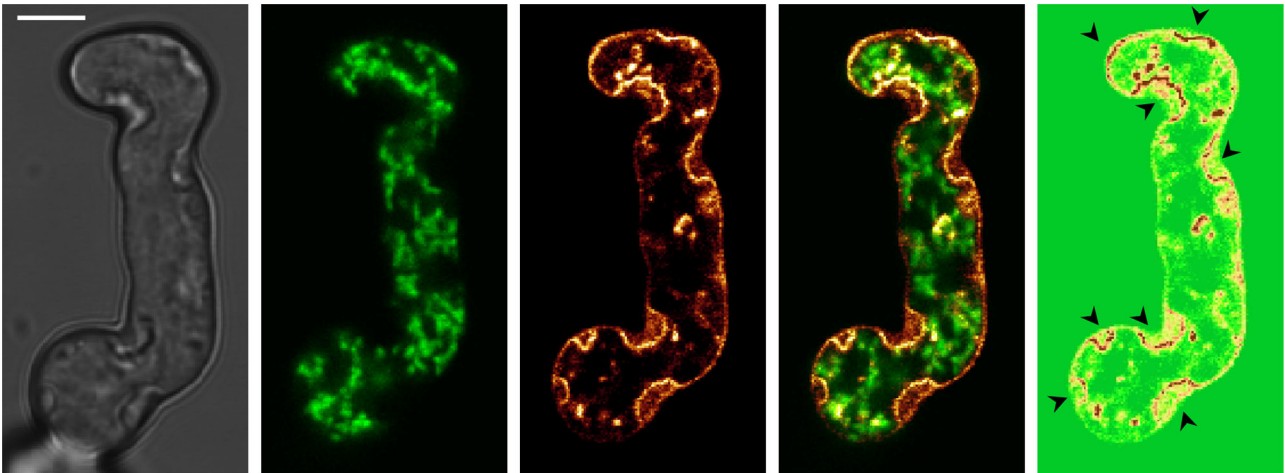

**Fig. 4** Multiple azole-induced cell wall carbohydrate patches invaginate the plasma membrane. *A. fumigatus* conidia expressing mitochondria-targeted red fluorescence protein (RFP) and GFP-tagged membrane-anchored Wsc1 were inoculated in Sabouraud medium. After 9 h incubation at 37 °C, medium was supplemented with 0.53 µg ml⁻¹ voriconazole. Fluorescence was analyzed with a confocal laser scanning microscope. The micrographs show bright field (left), fluorescence cross sections (green, RFP; glow dark color scheme, GFP) and an overlay of the two fluorescence cross sections of a representative hypha incubated for 4 h in the presence of voriconazole. The GFP fluorescence intensity is additionally visualized with a heatmap color scheme (right micrograph), the accumulation of Wsc1-GFP at sites of plasma membrane invaginations are indicated with arrow heads. The bar represents 5 µm

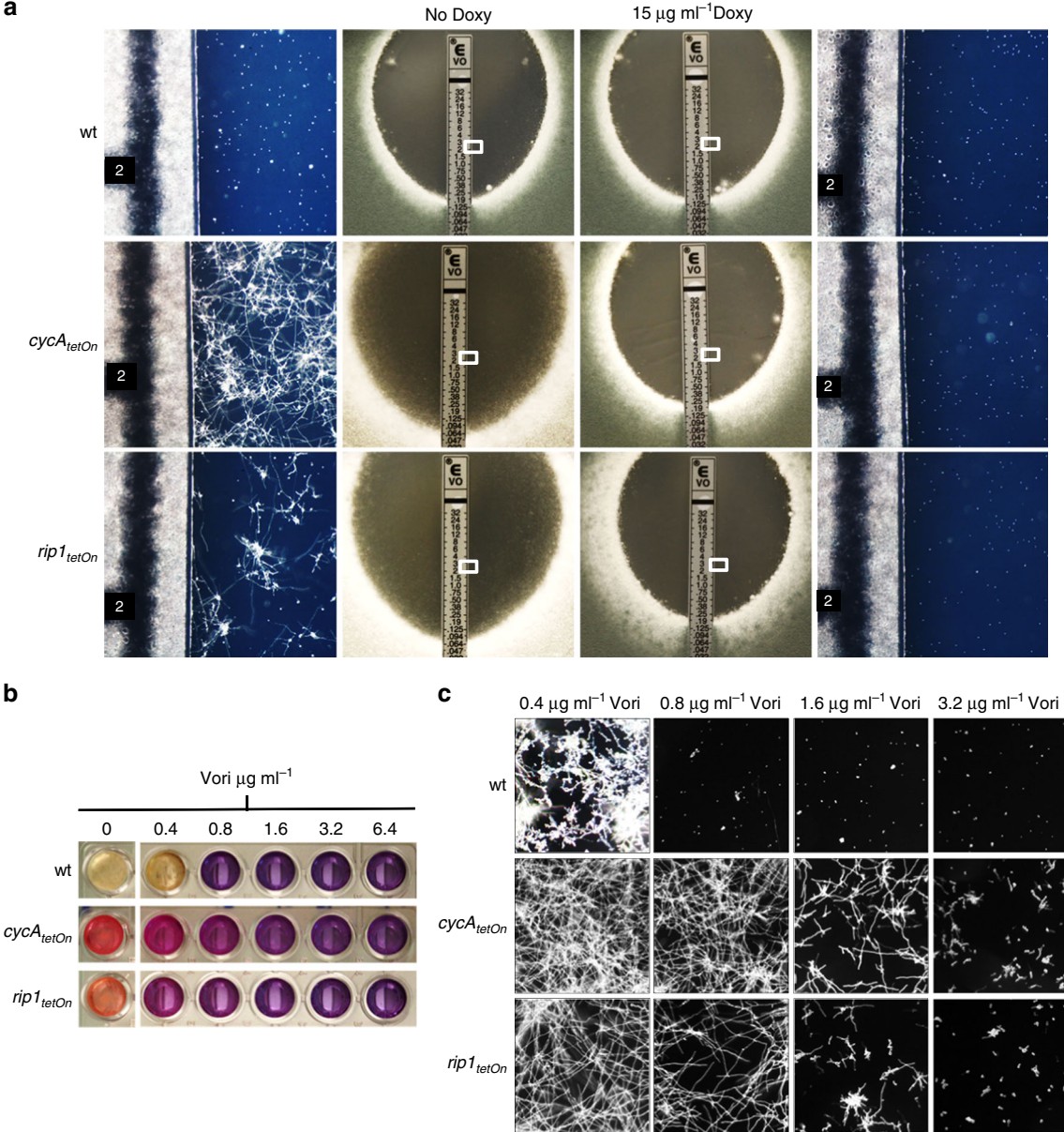

**Fig. 5** The fungicidal activity of azoles depends on the functionality of the conventional mitochondrial electron transport chain. **a** $1 \times 10^6$ conidia of wild-type (wt) and of the conditional cytochrome c ($cycA_{tetOn}$) and complex III ($rip1_{tetOn}$) mutants were spread on Sabouraud agar plates. When indicated, medium was additionally supplemented with $15\,\mu g\,ml^{-1}$ doxycycline to induce the conditional promoter ( + Doxy). Voriconazole Etest strips were applied and plates were incubated at 37 °C. Representative photos were taken after approximately 48 h. The panels next to the macroscopic Etest strip photos show magnifications of the framed sections. **b**, **c** Conidia of the indicated strains were inoculated in Sabouraud medium supplemented with resazurin (cell viability marker, $0.1\,\mu g\,ml^{-1}$) and the indicated amount of voriconazole and incubated at 37 °C. Macroscopic (**b**) and microscopic (**c**) images were taken after 42 h

chain. In contrast, the fungicidal activity seems to depend on the functionality of the conventional mitochondrial electron transport chain and can thus be clearly dissected from the fungistatic activity of voriconazole.

**Fungicidal activity of azoles is linked to patch formation**. The cell integrity loss following the vigorous invaginations of the plasma membrane suggests that the formation of the cell wall carbohydrate patches is responsible for the fungicidal activity of azole antifungals. We, therefore, questioned whether the shift of the antifungal activity of azoles from fungicidal to fungistatic observed with the conditional mutants affected in the cytochrome

respiratory pathway correlates with the formation of cell wall patches. As shown in Fig. 6, this is the case. The $cycA_{tetOn}$ and $rip1_{tetOn}$ mutants under repressed conditions exposed no carbohydrate patches at concentrations where patches are already observed in the wild-type (0.4, 0.8, and $1.6\,\mu g\,ml^{-1}$ voriconazole, Fig. 6). At these concentrations, voriconazole already exerts imposing fungistatic activity against $cycA_{tetOn}$ and $rip1_{tetOn}$ under repressed conditions. Higher azole concentrations (3.2 and $6.4\,\mu g\,ml^{-1}$ voriconazole) yielded significant patch formation, which also correlated with incremental inhibition of the minimal mycelial growth (Figs. 5c and 6).

We exploited the decoupling of the fungistatic and fungicidal activity of voriconazole observed with the $rip1_{tetOn}$ and $cycA_{tetOn}$

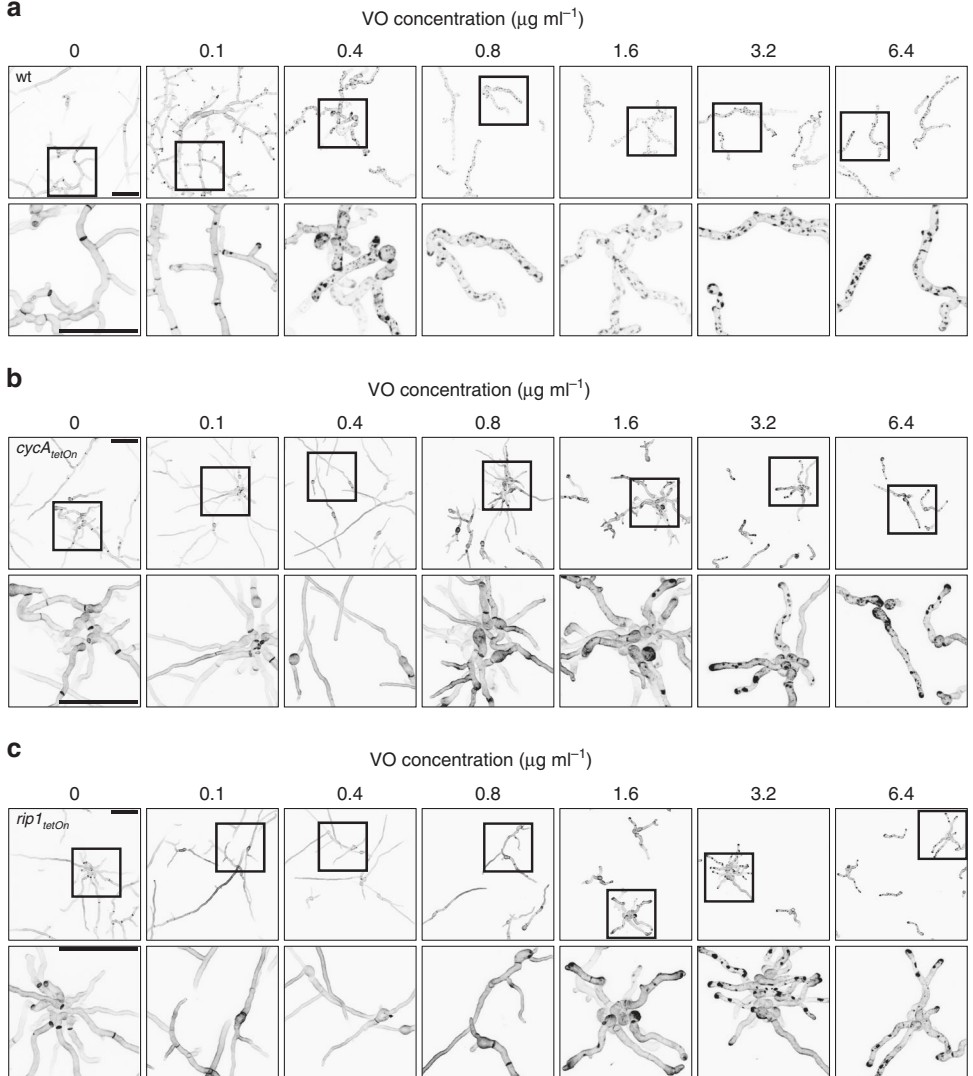

**Fig. 6** Fungistatic azole concentrations do not trigger cell wall carbohydrate patch formation in mutants affected in the conventional mitochondrial electron transport chain. **a**–**c** Conidia of wild-type (wt; **a**), cycA$_{tetOn}$ (**b**) and ript1$_{tetOn}$ (**c**) were inoculated in Sabouraud medium and incubated at 37 °C. After 10 h incubation at 37 °C, the medium was supplemented with the indicated amount of voriconazole. After additional 10 h incubation, hyphae were stained with calcofluor white, fixed and analyzed with a confocal laser scanning microscope. Depicted are representative images of optical stacks of the calcofluor white fluorescence (chitin) that cover the hyphae in focus. The lower panels show magnifications of the framed sections in the upper panels. Bars represent 50 μm and are applicable to all respective subpanels

mutants to investigate the relation of the patch formation and fungal cell death. Hyphae of the conditional mutants expressing mitochondria-targeted GFP were cultured under repressed conditions and then exposed to azole concentrations that induce the formation of cell wall patches. After azole co-incubation, hyphae were stained with calcofluor white to visualize the carbohydrate patches and subsequently analyzed with confocal laser scanning microscopy. Viable hyphal compartments were identified based on their tubular and dynamic mitochondrial morphology (Fig. 7a and Supplementary Movies 14 and 15). Depending on the azole concentration, a large number of the hyphal compartments were still alive after approximately 15–17 h azole exposure. Quantitative analysis of the hyphae revealed a significant and strong correlation of fungal cell death with the presence and size of the cell wall carbohydrate patches (Fig. 7b–d and Supplementary Fig. 2). Notably, especially at lower concentrations that, overall, were less fungicidal against the mutants under repressed conditions, we observed a small number

of compartments (< 4%) which were dead but exhibited no patches (Supplementary Fig. 2). This suggests that a minor fungicidal activity of azoles may exist independently from the fungicidal effect of the cell wall carbohydrate patch formation. Taken together, these data demonstrate that the cell wall patch formation greatly correlates with the fungicidal activity of the azoles, while the fungistatic azole concentrations do not necessarily yield patches.

**Inhibition of glucan synthesis delays fungicidal activity.** We speculated that inhibition of the formation of these patches would antagonize the fungicidal activity of voriconazole. As shown in Fig. 8a, inhibition of the β-1,3-glucan synthase Fks1 with the echinocandin caspofungin altered the microscopic appearance of the calcofluor white-stained cell wall carbohydrate patches that form in azole-treated *A. fumigatus* hyphae. The patches became much smaller and were more compact compared to the rather huge amorphous bodies under azole exposure alone.

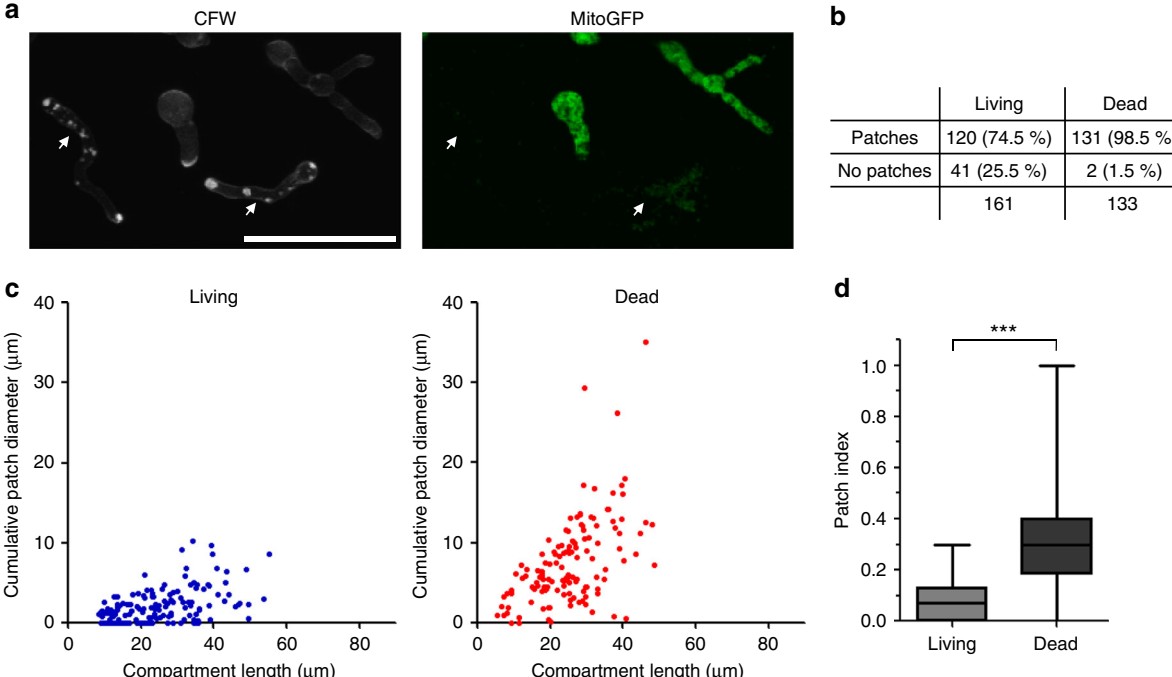

**Fig. 7** The presence and size of the cell wall carbohydrate patch correlate with death of individual hyphae. **a–d** Conidia of the conditional complex III mutant (*rip1*$_{tetOn}$) expressing mitochondria-targeted GFP were inoculated in Sabouraud medium under repressed conditions. After 10 h of incubation at 37 °C, medium was supplemented with 2.4 μg ml$^{-1}$ voriconazole and incubated for another 15 h. Hyphae were stained with calcofluor white, and analyzed with time-lapse laser scanning microscopy. **a** Representative images of dead and living hyphae. Left, calcofluor white fluorescence (patches), right, GFP fluorescence (mitochondria). Dead hyphae (arrows) were characterized by multiple cell wall carbohydrate patches, arrest of mitochondrial dynamics (compare Supplementary Movie 7), fragmentation of the tubular mitochondrial network and fading of the GFP fluorescence. Living hyphae show highly dynamic and tubular mitochondria (compare Supplementary Movie 7) and less or no cell wall carbohydrate patches. The bar indicates 50 μm. **b–c** Short time-lapse sequences of multiple hyphae were taken and each hyphal compartment was analyzed for viability, compartment length, and cumulative diameter of the containing cell wall carbohydrate patches. The depicted results are based on time-lapse microscopy data obtained from three technical replicates in one experiment. Two-hundred and ninety-four hyphal compartments were analyzed in total. Very similar results were obtained in two independent experiments with the *cycA*$_{tetOn}$ strain under similar conditions (Supplementary Fig. 1). **b** Absolute and relative numbers for living and dead compartments that exhibit patches or no patches. A significant number of living compartments exhibit no patches, while almost all dead compartments have patches. **c** The graphs indicate the cumulative patch diameter and compartment length for each living (blue) and dead (red) compartment. **d** Dead compartments exhibit a significantly higher ratio of the cumulative patch diameter and compartment length (patch index; depicted as box-and-whiskers graph). Statistical significance (***$p \leq 0.001$) was calculated with a Mann–Whitney test

Therefore, we analyzed the effect of β-1,3-glucan synthase inhibition on the survival of azole-treated *A. fumigatus* hyphae. Caspofungin was able to significantly increase the number of viable hyphal compartments after five and 6 h co-incubation with voriconazole (Fig. 8b). Very similar results were obtained using a conditional *fks1* mutant (*fks1*$_{tetOn}$[17]). Conditional repression of the β-1,3-glucan synthase gene *fks1* was able to significantly increase the number of microcolonies surviving 6 h voriconazole exposure (Fig. 8c). These data show that inhibition of cell wall patch accumulation delays the fungicidal effects of voriconazole.

**Azole-resistant clinical isolates do not form cell wall patches**. Azole antifungals are currently recommended as first-line therapy of invasive aspergillosis and other fungal infections. Unfortunately, the recent emergence of azole resistance in fungal pathogens challenges this approach and infections with azole-resistant *A. fumigatus* have been shown to result in up to 88% mortality[24,25]. If as our data suggest cell wall patches are related to the fungicidal activity of azoles, they should be absent in azole-resistant isolates. To test this hypothesis, three azole-susceptible and two azole-resistant clinical isolates were acquired from the National Reference Center for Invasive Fungal Infections. As shown in Fig. 9, the three azole-susceptible clinical isolates equally formed multiple cell-wall patches at low inhibitory

concentrations of voriconazole (0.8 μg ml$^{-1}$). Similarly, patches were also found after exposure to high azole concentrations (12.8 μg ml$^{-1}$). In sharp contrast, the azole-resistant clinical isolates did not form any patches after exposure to any of the tested azole concentrations. This clearly illustrates that, first, patches are also triggered in different *Aspergillus* isolates by azoles, and second, patch formation is linked to azole susceptibility.

## Discussion

Azoles exert a potent fungicidal activity against the major pathogen *A. fumigatus*. This activity is important for the clearance of life-threatening invasive fungal infections in the immunocompromised host. But the cellular mechanism responsible for the fungicidal activity remained unresolved so far. In this study, we provide new insights on how azole antifungals kill *A. fumigatus*. By constructing and studying a conditional CYP51 mutant, we have shown that the fungicidal activity of voriconazole is unambiguously linked to specific inhibition of the lanosterol 14α-demethylase. This inhibition results in suppression of hyphal growth within 1 h, the fungistatic effect. Importantly, hyphae remain fully viable at this point. The marked growth arrest is followed by extensive delivery of membrane protein-loaded transport vesicles to the cell membrane concomitant with excessive biogenesis of cell wall carbohydrates at defined spots

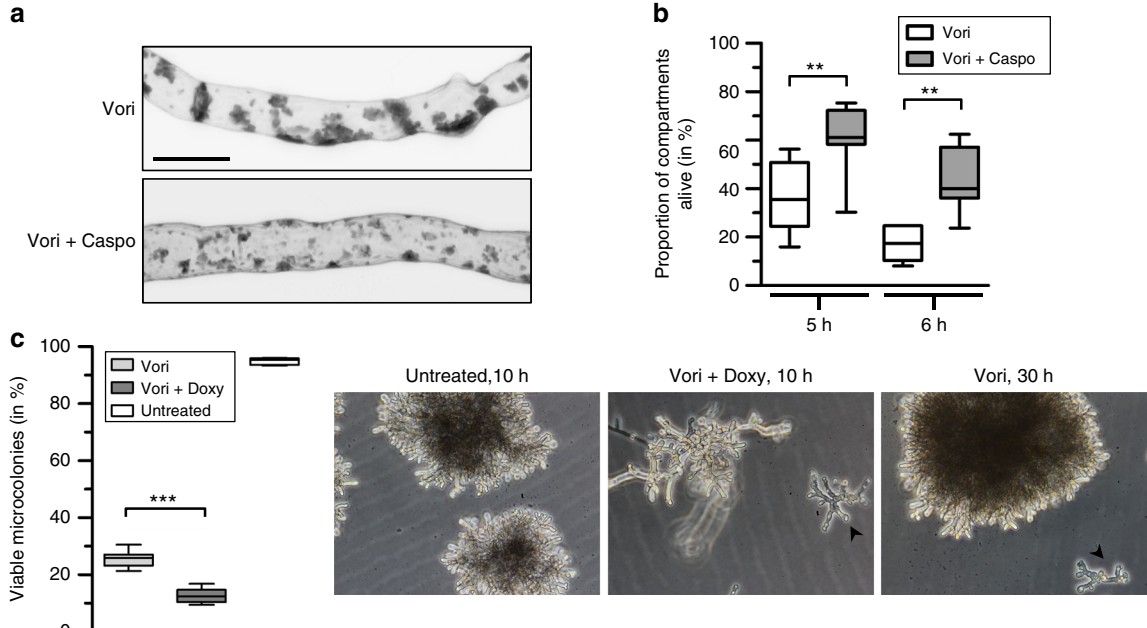

**Fig. 8** Inhibition of β-1,3-glucan synthesis attenuates the fungicidal activity of voriconazole. **a**, **b** Conidia of *A. fumigatus* wild-type (**a**) or wild-type expressing mitochondria-targeted GFP (**b**) were inoculated in Sabouraud medium on cover slips (**a**) or in an eight-well live cell microscopy slide (**b**) and incubated at 37 °C. After 9 h incubation, medium was supplemented with 1.27 µg ml$^{-1}$ voriconazole (Vori) and, when indicated, 30 min later additionally with 4 µg ml$^{-1}$ caspofungin (+ Caspo). **a** After 4 h co-incubation with antifungals, hyphae were fixed, stained with calcofluor white and analyzed with a confocal laser scanning microscope. Depicted are representative calcofluor white fluorescence images of optical stacks that cover the entire hypha in focus. Bar represents 10 µm and is applicable to both subpanels. **b** After 5 and 6 h co-incubation with antifungals, the mitochondrial morphology and dynamics of at least 975 hyphal compartments per condition were analyzed with a confocal laser scanning microscope in seven independent experiments. The box-and-whiskers graph shows the percentage of viable compartments under each condition. Statistical significance (**$p < 0.01$) was calculated with a two-tailed paired Student's *t*-test (assuming equal variances). **c** Conidia of the conditional β-1,3-glucan synthase mutant (*fks1*$_{tetOn}$) were inoculated in Sabouraud medium in 24-well plates and incubated at 37 °C under repressive conditions for 11 h. When indicated, media were subsequently supplemented with 1.27 µg ml$^{-1}$ voriconazole (Vori), or additionally with 10 µg ml$^{-1}$ doxycycline (Vori + Doxy) to induce expression of the β-1,3-glucan synthase Fks1. After additional 6 h incubation at 37 °C, medium was discarded. The wells were washed and supplemented with fresh medium without antifungals and doxycycline and the plates incubated for additional 10 to 30 h at 37 °C. The percentage of surviving microcolonies was microscopically determined and shown in the depicted box-and-whiskers graph. Criteria for viability were continuation of growth combined with light refraction; viable hyphae were bright, and dead hyphae were dark. Exemplary bright-field image of dead and viable microcolonies as assessed after the indicated additional incubation time are shown on the right, dead microcolonies are indicated with arrow heads. The depicted experimental results are representative of four independent blinded experiments under similar conditions. Statistical significance (***$p \leq 0.001$) was calculated with a two-tailed unpaired (assuming equal variances) Student's *t*-test

along the hyphae. The fungus fails to underpin the bolstered cell wall synthesis with hyphal protrusions. Instead, the excessive accumulation of cell wall carbohydrates bends the membrane to the inside. This results in tremendous cell wall stress as exemplified by the strong induction of the cell wall salvage system. Finally, the cell wall carbohydrate patches impale the plasma membrane, thereby causing sudden cell integrity failure and death of the fungus. Hyphal septa improve the fungal survival by sealing off damaged compartments from the viable mycelium, but cannot fully prevent the profound fungicidal activity, which takes effect in parallel in all hyphal compartments. Our model, summarized in Fig. 10, is well supported by our additional findings: First, in the azole-tolerant *cycA*$_{tetOn}$ and *rip1*$_{tetOn}$ mutants the cell wall carbohydrate patches are predominantly found in dead and not in viable compartments of azole-inhibited hyphae. Second, inhibition of glucan synthesis can attenuate the formation of cell wall carbohydrate patches and delay the fungicidal effect of lanosterol 14α-demethylase inhibition in *A. fumigatus*.

A review of the literature revealed that our results are in good agreement with results reported and discussed more than 30 years ago[6]. At this time, Kerkenaar and Barug observed that the filamentous fungi *Ustilago maydis* and *Penicillium italicum* form

chitin patches after treatment with the azole antifungal imazalil and the morpholine fenpropimorph[26]. Similar, irregular chitin deposits were found in the yeast form and filaments of the pathogenic yeast *Candida albicans*[6,27]. Kerkenaar, Barug, and Bossche hypothesized that the occurrence of chitin deposits may differentially affect growth and viability, depending on the morphotype and cell wall composition of the species[6,26]. However, the exact mechanism that kills filamentous fungi was not further elaborated. Our results clearly demonstrate the role of cell wall carbohydrates, especially of β-1,3-glucan, for the fungicidal activity of azoles.

Our time-lapse microscopy results revealed that growth inhibition occurs in the first hour after azole exposure while cell wall carbohydrate patches need more than 2 h to manifest. This indicates that (1) the initial fungistatic effect of CYP51 inhibition does not result from the cell wall carbohydrate patches, and (2) that the formation of cell wall carbohydrate patches is a secondary event that results from or occurs in parallel to the arrest of growth. This model is greatly supported by the distinct azole susceptibility phenotype of the *rip1*$_{tetOn}$ and *cycA*$_{tetOn}$ mutants. At lower effective azole concentrations these mutants do not form patches but are strikingly inhibited in growth. In contrast, higher

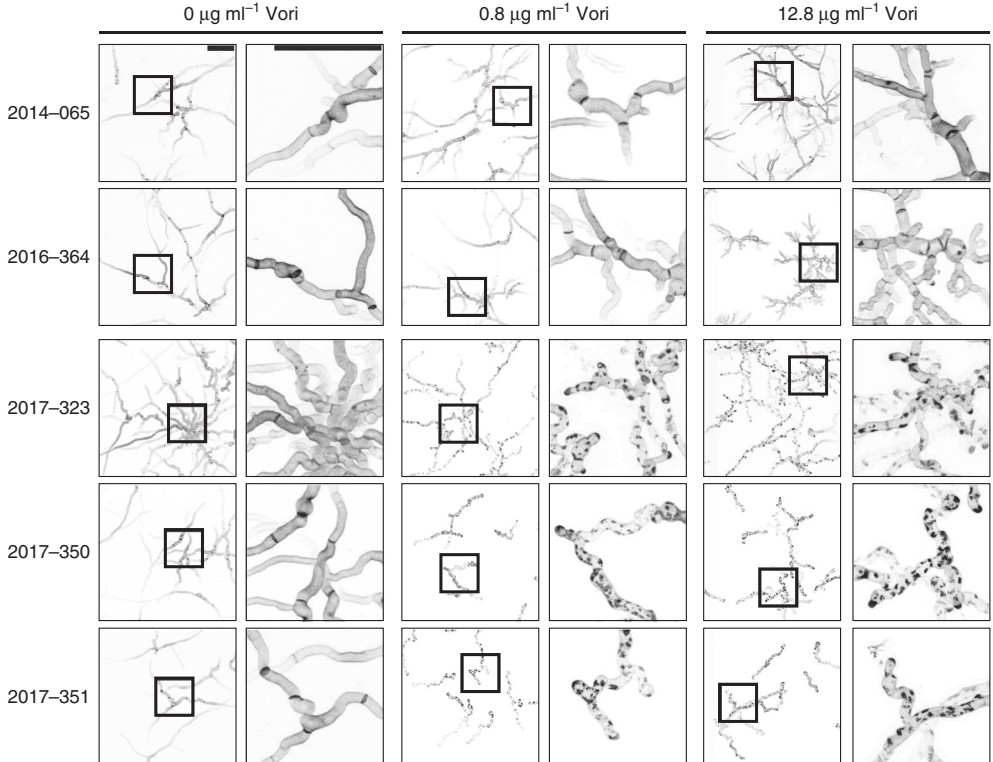

**Fig. 9** Azoles trigger patch formation in azole-susceptible but not in azole-resistant *A. fumigatus* clinical isolates. Conidia of three azole-susceptible (2017–323, 2017–350, 2017–351) and two azole-resistant (2014–065, 2016–364) clinical isolates of *A. fumigatus* were inoculated in Sabouraud medium. After 10 h incubation at 37 °C, medium was supplemented with the indicated amount of voriconazole. After additional 10 h incubation, hyphae were stained with calcofluor white, fixed and analyzed with a confocal laser scanning microscope. Depicted are representative images of optical stacks of the calcofluor white fluorescence (chitin) that cover the hyphae in focus. The right panels show magnifications of the framed sections in the panels shown on the left. Bars represent 50 μm and are applicable to all respective subpanels

azole concentrations additionally induce patch formation, which then correlates with death of the fungus. It remains speculative whether one of the many other previously proposed antifungal activities of azole derivatives is responsible for the initial fungistatic effect[6–10].

Although not the intentional focus of our study, our results might be interpreted to suggest that inhibition of β-1,3-glucan synthesis could antagonize the fungicidal activity of azole antifungals. This could have important implications because combinatory therapy is currently under consideration to improve the poor outcome of invasive fungal infections. However, it has to be noted, that our observations require a well-defined time course of azole–echinocandin application unlikely to occur in vivo. Furthermore, our readout was solely focused on the fungicidal activity but not on the fungistatic effect and it is important to note that inhibition of glucan synthesis can significantly delay but not fully block azole-induced hyphal death. Several factors can account for this, including excessive synthesis of other major cell wall carbohydrates, e.g., chitin[17], α-1,3-glucan or galactomannan, which also results in patches that challenge the cell wall integrity. Thus, our data do not contradict previous in vitro studies suggesting synergistic activity of azoles and echinocandins[28,29] as well as a recent randomized clinical trial, which suggested potential beneficial effects of combination therapy[30].

Finally, our results open up new approaches to evaluate the efficacy of antifungal therapy. Refractory or progressive invasive aspergillosis may occur and often it remains unclear whether the azole-based antifungal therapy is effective or not. Reported reasons for non-effective therapy and breakthrough invasive fungal infections are resistance due to mutations in CYP51, upregulation

of CYP51 expression or upregulation of efflux pumps or, as a matter of debate, antifungal tolerance or host-specific insufficient bioavailability of the drug[24,31,32]. In this study, we could demonstrate that azole-susceptible clinical *A. fumigatus* isolates regularly exhibit cell wall carbohydrate patches upon exposure to inhibitory voriconazole concentrations. Azole-resistant clinical isolates, however, did not form cell wall carbohydrate patches. This could be exploited to establish novel protocols for more rapid in vitro susceptibility testing of clinical isolates in routine diagnostics. Similar, evidence of chitin or glucan patches within hyphae in host specimens would indicate that the administered azole successfully inhibited CYP51. In contrast, the absence of any chitin or glucan patches could indicate that CYP51 was not effectively inhibited. To substantiate these diagnostic approaches and the general applicability to other fungal species, additional studies are required.

## Methods
**Strains and culture conditions**. The non-homologous end joining-deficient *A. fumigatus* strain AfS35 (ref. [33]), a derivative of D141, was used as wild-type in this study. The Δ*rho4* mutant and the complemented Δ*rho4* + *rho4* mutant (*rho4*) were described previously[18]. To conditionally express CYP51, a doxycycline-inducible promoter system (pkiA-tetOn; pYZ002) was inserted before the coding sequence of *cyp51A*, essentially as described before[21]. Subsequently, *cyp51B* was replaced with a self-excising hygromycin B resistance cassette (pSK485[34]), thereby yielding *cyp51A*$_{tetOn}$Δ*cyp51B*. To conditionally express *cycA* and *rip1*, the doxycycline-inducible promoter system (pkiA-tetOn; pYZ002) was inserted before the coding sequence of the respective genes. To visualize mitochondria with mitochondria-targeted GFP or RFP, the respective strains were transformed with the construct pCH005 or pYZ012[21]. The conditional *fks1*$_{tetOn}$ strain and the strains that constitutively express cytosolic GFP, Wsc1-GFP or MidA-GFP were described previously[17,18]. To construct the cell wall salvage reporter, the sequence of the

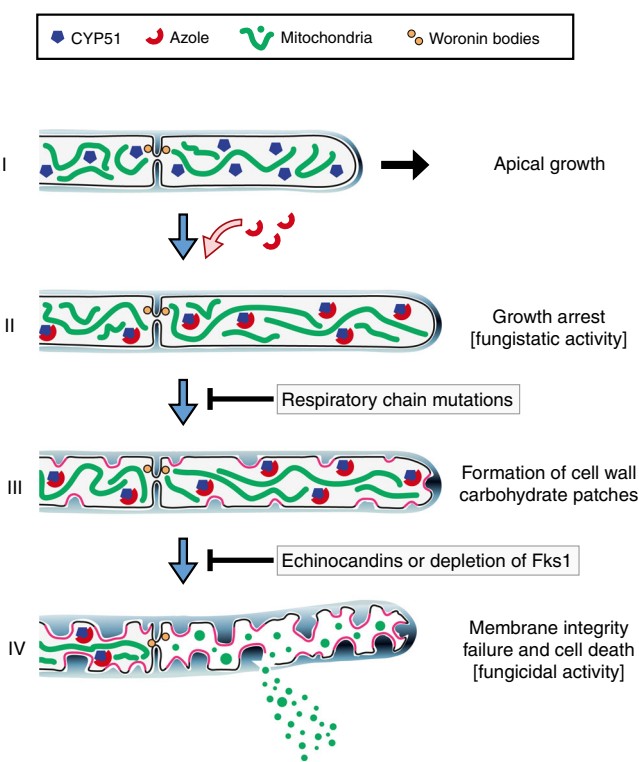

**Fig. 10** Proposed model for the fungicidal activity of azole antifungals against *A. fumigatus*. Azole enters the growing *Aspergillus* hypha (I) and binds to the target enzyme lanosterol 14α-demethylase (CYP51). The inhibition of CYP51 results in the depletion of ergosterol and suppression of growth in less than 1 h (II). This is followed by extensive delivery of membrane protein-loaded transport vesicles to the cell membrane (accentuation in pink) and the induction of excessive β-1,3-glucan and chitin synthesis at defined spots along the hyphae (III). The continually increasing cell wall carbohydrate patches vigorously invaginate the plasma membrane, eventually resulting in membrane integrity failure and death of hyphae. Septa adjacent to damaged compartments are sealed off by Woronin bodies, specialized organelles that plug the septal pores, and improve survival of azole-exposed hyphae (IV). Inhibition of β-1,3-glucan synthesis, either by echinocandin antifungals or by reduced expression of the β-1,3-glucan synthase Fks1, attenuates the patch formation and delays the fungicidal activity of the azole. Mutants with a dysfunctional respiratory chain are strongly inhibited in growth but do not form cell wall patches, and thus do not die in the presence of lower inhibitory concentrations of azoles

*Aspergillus niger agsA* promoter fused to the coding sequence of the firefly luciferase of pNB04[16] was PCR-amplified and cloned into the *Pst*I and *Pme*I sites of pSK379[35], thereby yielding pBG005. The ptrA resistance cassette of pBG005 was subsequently replaced with a phleomycin resistance cassette. The resulting plasmid, pBG005-phleo, was transformed in the AfS35 wild-type strain. All experiments were performed in Sabouraud medium [4% (w/v) D-glucose, 1% (w/v) peptone (#LP0034; Thermo Fisher Scientific; Rockford, IL, US), pH 7.0]. Doxycycline was purchased from Clontech (#631311; Mountain View, CA, USA). Resazurin (#R7017), calcofluor white (#F3543), aniline blue diammonium salt (#415049), trypan blue (#6146), and caspofungin diacetate (#SML0425) were obtained from Sigma–Aldrich (St. Louis, MO, USA). Voriconazole was purchased from Apexbt Technology LLC (#A4320; Houston, TX, USA). Luciferin was purchased from Promega (#E1601; Fitchburg, WI, USA). Etest strips were purchased from bio-Mérieux (Marcyl'Etoile, France).

**Microscopy**. Confocal laser scanning microscopy was performed with a Leica SP5 microscope (Leica Microsystems; Mannheim, Germany) equipped with a temperature-controllable environment chamber. For live cell microscopy, conidia were inoculated in 15 μ-Slide eight-well (#80826) slides or 60 μ-Dish (#81156) dishes (Ibidi; Martinsried, Germany). When indicated, samples were fixed with 3.7% formaldehyde in Dulbecco's phosphate-buffered saline for 3 min. Chitin and glucan were stained with calcofluor white and aniline blue, respectively. For calcofluor white-staining, fixed samples were stained with 10 mg ml$^{-1}$ calcofluor white dissolved in ddH2O for approximately 1 min and unfixed samples were stained by supplementing the medium with 3.33 μg ml$^{-1}$ calcofluor white for at least 5 min. Staining procedures for aniline blue were described previously[36]. Fixed calcofluor white-stained samples were mounted with Vectashield mounting medium (H-1000; Vector, Burlingame, CA, USA). Fluorescence microscopy of aniline blue-stained samples was performed with a BX61 microscope (Olympus, Tokyo, Japan) and a modified filter cube as described recently[36]. An Axiovert 25 inverted microscope (Carl Zeiss MicroImaging, Göttingen, Germany) and an EOS 550D digital camera (Canon, Tokyo, Japan) were used for samples that were examined with bright field microscopy only. Samples for blinded experiments (see figure legends) were allocated randomly by an investigator. The data were subsequently collected by another investigator without knowing the allocation of the samples or groups. The Shapiro–Wilk test results were used to assume equal variances.

**Cell wall salvage reporter assay**. Conidia were inoculated in white 96-well polystyrene microplates with transparent bottom (#655095, #656171) purchased from Greiner Bio-One (Kremsmünster, Austria). After 7 h incubation at 37 °C, medium was supplemented with the indicated amount of voriconazole and 0.5 mM luciferin. Luminescence was measured over time at 37 °C with a Clariostar microplate reader obtained from BMG Labtech (Ortenberg, Germany).

**Data availability**. The data that support the findings of this study are available in this article and its Supplementary Information files, or from the corresponding author upon request.

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

## Acknowledgements

This work was supported by the German Research Foundation (DFG–WA 3016/2-1 and DFG–WA 3802/1-1) and the Förderprogramm für Forschung und Lehre (FöFoLe) of the Medical Faculty of the Ludwig-Maximilians-Universität München. Work in the NRZMyk was supported by the Robert Koch-Institut from funds of the Federal Minitry for Health (grant-No.: 1369–240) and by the Federal Ministry for Education and Science within the Program InfectControl 2020 (project FINAR grant No.: 03ZZ0809A).

## Author contributions

J.W. conceived the study; B.G., V.L., I.K., N.W., O.K., C.H., and J.W. designed the experiments; B.G., V.L., I.K., Z.Z., C.H., and J.W. performed the experiments; O.K. contributed the clinical *A. fumigatus* isolates; C.H. contributed the cell wall salvage reporter system; all authors analyzed and discussed the data; B.G., V.L., I.K., O.K., and J. W. wrote the manuscript.
