## [Peer Review File · Nature Communications]

Reviewers' comments:

Reviewer #1 (Remarks to the Author):

This is an interesting, exciting, and well-written paper that reveals the mechanism by which the azole antifungals exert their fungicidal activity against the important fungal pathogen, *Aspergillus fumigatus*. The azoles inhibit sterol demethylase and thereby interfere with sterol biosynthesis, ultimately leading to reduced ergosterol content in the fungal cell membrane. However, the mechanism by which they inhibit growth appears to involve the accumulation of toxic sterol intermediates in the fungal cell membrane. Importantly, while the azoles are fungistatic against yeast, they are fungicidal against *Aspergillus fumigatus*. Of note, voriconazole is front-line therapy for invasive aspergillosis. Understanding how this class of antifungal elicits this species-specific fungicidal effect is of importance and represents a significant unanswered question in antifungal pharmacology. Using very elegant microscopy experiments, these investigators have shown that the azole antifungals induce the cell wall salvage pathway in *A. fumigatus* correlating with loss of cell integrity and hyphal compartment death. Collectively these data indicate that the azoles result in synthesis of cell wall carbohydrate patches that penetrate the plasma membrane resulting in fungal death.

The manuscript is written very well and is easy to follow. The introduction clearly articulates the question being addressed and the rationale for these studies. Results are clearly communicated and figures and supplemental data are very well-presented. Methods are sufficiently detailed and clear. Finally, the discussion represents a fair and appropriate treatment of the findings. The authors are cautious not to over-interpret their results. Moreover, each question that I noted in the margins during my initial reading of this manuscript were addressed in the discussion. In short, this is an excellent paper that communicates new knowledge of sufficient importance and interest to warrant publication in *Nature Communications* in my opinion. Furthermore, I can make no reasonable suggestions for improvement of this manuscript.

Reviewer #2 (Remarks to the Author):

This paper aims to address the mechanisms that mediate the antifungal effects of azoles on the filamentous ascomycete *Aspergillus fumigatus*. Azoles are well-known inhibitors of ergosterol biosynthesis. Ergosterol plays a key role in the physiology of cellular membranes by forming lipid domains, together with sphingolipids. Therefore ergosterol has multiple tasks at all levels of intracellular traffic. It is generally admitted that lipid rafts are crucial for the physiology of the Golgi, for example for the delivery of secretory vesicles to the plasma membrane. As this exocytic traffic transports cell wall-modifying enzymes to the plasma membrane, it should come as no surprise that targeting ergosterol biosynthesis results both in major polarity defects and in delocalization/dysfunction of the cell wall biosynthetic machinery that, inevitably, results in activation of the CWI pathway. The gallery of morphologically abnormal germlings, the formation of cell wall patches and the increased septation phenotypes that the authors display in their figures is not different from images depicting mutants affected in different steps of exocytosis that exist in the literature. Therefore I conclude that in its present form this manuscript is highly descriptive and does not go any deep into the actual cellular/molecular steps that result in the observed phenotypes. The observation of synthetic interactions between azoles and mitochondrial proteins are potentially interesting but have not been pursued beyond its mere description. The conclusion that the title reflects is not justified, the formation of patches correlates with death, but none of the experiments supports the far-fetched contention that they are the actual cause of death. In summary, the paper

belongs to a very specialized microbiology journal and is very far of reaching the levels of impact, novelty and mechanistic insight that one would expect to find in Nat Comm.

Reviewer #1

This is an interesting, exciting, and well-written paper that reveals the mechanism by which the azole antifungals exert their fungicidal activity against the important fungal pathogen, *Aspergillus fumigatus*. The azoles inhibit sterol demethylase and thereby interfere with sterol biosynthesis, ultimately leading to reduced ergosterol content in the fungal cell membrane. However, the mechanism by which they inhibit growth appears to involve the accumulation of toxic sterol intermediates in the fungal cell membrane. Importantly, while the azoles are fungistatic against yeast, they are fungicidal against *Aspergillus fumigatus*. Of note, voriconazole is front-line therapy for invasive aspergillosis. Understanding how this class of antifungal elicits this species-specific fungicidal effect is of importance and represents a significant unanswered question in antifungal pharmacology. Using very elegant microscopy experiments, these investigators have shown that the azole antifungals induce the cell wall salvage pathway in *A. fumigatus* correlating with loss of cell integrity and hyphal compartment death. Collectively these data indicate that the azoles result in synthesis of cell wall carbohydrate patches that penetrate the plasma membrane resulting in fungal death.

The manuscript is written very well and is easy to follow. The introduction clearly articulates the question being addressed and the rationale for these studies. Results are clearly communicated and figures and supplemental data are very well-presented. Methods are sufficiently detailed and clear. Finally, the discussion represents a fair and appropriate treatment of the findings. The authors are cautious not to over-interpret their results. Moreover, each question that I noted in the margins during my initial reading of this manuscript were addressed in the discussion. In short, this is an excellent paper that communicates new knowledge of sufficient importance and interest to warrant publication in *Nature Communications* in my opinion. Furthermore, I can make no reasonable suggestions for improvement of this manuscript.

We are extremely grateful for the very positive assessment of our manuscript by this Reviewer. The positive evaluation by this reviewer also encouraged us to avoid massive changes to the overall flow of the manuscript while at the same time addressing the critical remarks made by Reviewer #2.

Reviewer #2

This paper aims to address the mechanisms that mediate the antifungal effects of azoles on the filamentous ascomycete *Aspergillus fumigatus*. Azoles are well-known inhibitors of

ergosterol biosynthesis. Ergosterol plays a key role in the physiology of cellular membranes by forming lipid domains, together with sphingolipids. Therefore ergosterol has multiple tasks at all levels of intracellular traffic. It is generally admitted that lipid rafts are crucial for the physiology of the Golgi, for example for the delivery of secretory vesicles to the plasma membrane. As this exocytic traffic transports cell wall-modifying enzymes to the plasma membrane, it should come as no surprise that targeting ergosterol biosynthesis results both in major polarity defects and in delocalization/dysfunction of the cell wall biosynthetic machinery that, inevitably, results in activation of the CWI pathway. The gallery of morphologically abnormal germlings, the formation of cell wall patches and the increased septation phenotypes that the authors display in their figures is not different from images depicting mutants affected in different steps of exocytosis that exist in the literature. Therefore I conclude that in its present form this manuscript is highly descriptive and does not go any deep into the actual cellular/molecular steps that result in the observed phenotypes.

We feel that the impression that our manuscript is “descriptive” and findings are “no surprise” may accidentally have been caused by our attempt to present the new findings in a way that is easily accessible for educated non-expert readers to account for the broad audience of Nature Communications. To avoid this impression, we have now emphasized more clearly the medical importance of *A. fumigatus* and the crucial role of azole antifungals to combat infections caused by this pathogen (Introduction page 5, lines 88-93). Furthermore, we thoroughly revised the discussion of our manuscript to improve the reasoning and put emphasis on the novelty of our work (Discussion, page 15ff, lines 318-342 and new Figure 10 which summarizes our model). Our findings allow for the first time a clear dissection of fungistatic versus fungicidal effects of azoles and provide a robust explanation for the different action not only against susceptible versus resistant *A. fumigatus* isolates but potentially also for other fungal pathogens. Finally, we have tried to underline the translational potential of our findings in generating innovative susceptibility testing approaches (Discussion, page 18, lines 379-397).

The observation of synthetic interactions between azoles and mitochondrial proteins are potentially interesting but have not been pursued beyond its mere description.

While we fully agree with reviewer #2, that more can be learned about mitochondrial function in *Aspergillus* by further dissecting the findings of our study, we have decided to focus on the results, that directly impact the main line of arguments in our manuscript. In this respect, it is

a novel and important finding, that disruption of the mitochondrial respiratory chain allows a dissection of the fungistatic from the fungicidal effect of azole antifungals.

The conclusion that the title reflects is not justified, the formation of patches correlates with death, but none of the experiments supports the far-fetched contention that they are the actual cause of death. In summary, the paper belongs to a very specialized microbiology journal and is very far of reaching the levels of impact, novelty and mechanistic insight that one would expect to find in Nat Comm.

To avoid the impression of coincidence rather than causal relationship, we have tried to emphasize the following arguments more precisely:

- i) Fungistatic and fungicidal effects of voriconazole can be dissected
- ii) Formation of cell wall patches only occurs during fungicidal action of voriconazole
- iii) Fungicidal effects of voriconazole are associated with rapid membrane ruptures releasing cytoplasmic content
- iv) Formation of beta-D-glucan by Fks1 in the cell wall patches is essential for the efficient fungicidal activity of voriconazole

Together with Reviewer #1, we feel that this is a strong line of evidence supporting our conclusions. To further illustrate this, we have designed a new supplemental figure that summarizes our novel model of how azoles kill the important fungal pathogen *A. fumigatus* (Figure 10).